# Recent Advances in Aptasensor for Cytokine Detection: A Review

**DOI:** 10.3390/s21248491

**Published:** 2021-12-20

**Authors:** Jinmyeong Kim, Seungwoo Noh, Jeong Ah Park, Sang-Chan Park, Seong Jun Park, Jin-Ho Lee, Jae-Hyuk Ahn, Taek Lee

**Affiliations:** 1Department of Chemical Engineering, Kwangwoon University, 20 Kwangwoon-Ro, Nowon-gu, Seoul 01897, Korea; jkim629@sogang.ac.kr (J.K.); nsw26510@kw.ac.kr (S.N.); m3m33@kw.ac.kr (J.A.P.); 2Department of Electronics Engineering, Chungnam National University, 99 Yuseong-gu, Daejeon 34134, Korea; psc701@o.cnu.ac.kr; 3Department of Electrical Engineering, Kwangwoon University, 20 Kwangwoon-Ro, Nowon-gu, Seoul 01897, Korea; sjnu96@kw.ac.kr; 4School of Biomedical Convergence Engineering, Pusan National University, 49 Busandaehak-ro, Yangsan 50612, Korea; leejh@pusan.ac.kr

**Keywords:** cytokine, aptamer, aptasensor, electrochemical biosensor, electrical biosensor

## Abstract

Cytokines are proteins secreted by immune cells. They promote cell signal transduction and are involved in cell replication, death, and recovery. Cytokines are immune modulators, but their excessive secretion causes uncontrolled inflammation that attacks normal cells. Considering the properties of cytokines, monitoring the secretion of cytokines in vivo is of great value for medical and biological research. In this review, we offer a report on recent studies for cytokine detection, especially studies on aptasensors using aptamers. Aptamers are single strand nucleic acids that form a stable three-dimensional structure and have been receiving attention due to various characteristics such as simple production methods, low molecular weight, and ease of modification while performing a physiological role similar to antibodies.

## 1. Introduction

When external pathogens invade the body, immune cells release low-molecular-weight proteins, such as cytokines, to regulate the inflammatory response and homeostasis [1]. Cytokines are protein immune regulators secreted by immune cells, and their molecular weights range from approximately 6 to 70 kDa [2]. They facilitate cell-to-cell communication and act on cytokine-secreting hair cells underlying neighboring cells. Cytokines also play important roles in cell replication and apoptosis [3], cancer [4], damaged tissue repair, and other physiological functions [1]. Some cytokines have been developed and reported as therapeutic agents and immune regulators [5,6]. Considering that cytokines are important indicators of physical health, monitoring them is of tremendous value in medicine and biological research [7].

Cytokines are immune regulators against external pathogens, but excessive cytokine secretion causes uncontrolled inflammatory reactions, resulting in attacks on normal cells [8,9]. Moreover, secondary infection symptoms occur as the DNA of attacked normal cells is altered. This phenomenon is called a “cytokine storm”. The cytokine storm can even kill healthy individuals without underlying diseases [8,9]. It has been identified as a major cause of high mortality in patients with avian flu [10], Spanish flu [11], and severe acute respiratory syndrome (SARS) [12]. In addition, those infected with Middle East respiratory syndrome (MERS) [13], an acute respiratory infectious disease prevalent in the Middle East in 2012, deteriorated without an underlying disease [14]. The medical community has recognized a cytokine storm as the cause of this phenomenon [10,11,12,13,14]. In December 2019, the first case of coronavirus disease 2019 (COVID-19) caused by the severe acute respiratory syndrome coronavirus 2 (SARS-CoV-2) occurred in Wuhan, China, causing a global pandemic [15]. SARS-CoV-2 shares many genomic and structural similarities with SARS- and MERS-associated coronaviruses [16]. It has been reported that a cytokine storm is instrumental in the pathogenesis of COVID-19 [17]. Early diagnosis of cytokine storms through monitoring of cytokine levels can prevent progression to serious diseases and achieve better outcomes. Therefore, rapid and accurate detection of cytokines is essential.

An aptamer is a single-stranded nucleic acid with a nucleotide sequence of approximately 15–40 mer; it has a stable tertiary structure and can bind to a target molecule with high affinity and specificity [18,19]. Aptamers are discovered using the systemic evolution of ligands by exponential enrichment (SELEX) published by the Lary Gold group in 1990 [20]. SELEX selectively acquires target-bound nucleic acid sequences after mixing a random nucleic acid library with target molecules. The sequences in the nucleic acid library are composed of a primer-binding platform with two identical sequences at both ends and a random sequence in the middle. SELEX selectively amplifies oligonucleotide sequences bound to a target molecule by incubating a nucleic acid library with a target molecule via polymerase chain reaction (PCR) or reverse transcription-polymerase chain reaction (RT-PCR). Aptamers with high affinity and specificity for the target molecule are developed by repeated incubation of the target molecule with the amplified oligonucleotide sequence [21].

The high specificity and affinity of aptamers for binding to target molecules are being compared to those of antibodies. Thus, aptamers are expected to replace antibodies owing to the following advantages. Aptamers have smaller molecular weights than antibodies, rendering them easier to modify. Moreover, because aptamers are chemically manufactured in mass production, their production cost is lower than that of antibodies [22]. Furthermore, aptamers are not recognized as antigens by the immune system [23]. Thus, in view of these unique properties of aptamers, research on their application in diagnosis, therapeutics, and drug delivery is actively being conducted [24,25].

A biosensor is a device that detects a target molecule using a probe derived from a biomolecule and then identifies physical changes due to the interaction between the bio-probe and the target molecule [26]. This target molecule recognition process is a key factor in sensor performance. An immunosensor that incorporates an antibody as a bio-probe is currently the most commonly used biosensor because of the high specificity and affinity between the antibody and the target molecule. However, it is sensitive to pH and temperature, has a short shelf-life, and is dependent on the time-consuming process of new antibody development [27,28]. An aptasensor is a biosensor that detects a target molecule using an aptamer as a bio-probe. It is expected to be an alternative to solve the problems associated with antibody-based biosensors by exploiting the advantages of aptamers described above. However, there are still many challenges to overcome in the commercialization of aptasensors owing to their low specificity and affinity compared to antibodies. This review summarizes the current research on methods to improve the performance of aptasensors and biosensors for detecting cytokines.

## 2. Electrochemical-Based Detection

### 2.1. Electrochemical Biosensor

Electrochemical biosensors have been used in various fields to detect analytes via immunoassays [29,30,31]. Electrochemical biosensors are gaining attention because of their low cost, sensitivity, device miniaturization, portability, and relatively fast response time [32,33]. The main advantage of electrochemical technology is its high sensitivity and inexpensive, electrical signal-based equipment [34,35,36,37]. Therefore, the development of electrochemical biosensors has increased dramatically in recent years. In addition, platforms combining an aptasensor [36,37] and electrochemical technology can be widely applied to health monitoring [38], clinical [39] and medical diagnoses [40], and point-of-care diagnostic tests.

Electrochemistry is the study of electricity generated by the movement of electrons between redox species. An electrochemical biosensor detects an analyte through the electrical signal generated when a redox reaction occurs on the electrode surface [41,42]. Cyclic voltammetry (CV) [43], normal pulse voltammetry (NPV) [44], differential pulse voltammetry (DPV) [45], and square wave voltammetry (SWV) [46] are divided according to the pulse waveform to which the voltage is applied. In addition, electrochemical impedance spectroscopy (EIS) is an effective electrochemical analysis method for measuring the impedance of an electrode surface according to frequency changes. Figure 1 shows a simple schematic of the electrochemical measurements. In this section, an electrochemical-based aptasensor is introduced.

### 2.2. Voltametry

CV is the most commonly used electrochemical technique for investigating the characteristics of redox reactions. This technique measures the current generated by applying a triangular waveform voltage to the working electrode over time [45]. A recent study by Lee group proposed an electrochemical biosensor composed of a multifunctional DNA four-way junction for the detection of tumor necrosis factor (TNF)-α [47]. Figure 2a shows a schematic of the fabricated biosensor. The DNA four-way junction consists of an aptamer sequence that recognizes TNF-α and a region with a mismatched sequence into which Ag ions can be intercalated. Ag ions were subjected to CV for evaluating the performance of the biosensor fabricated as a redox-active species. Redox peaks of Ag ions were found in the CV curve, and the concentration of TNF-α was quantified based on the intensity of this signal. The proposed electrochemical biosensor was able to detect effectively in the concentration range of 0.15 pg/mL to 150 ng/mL (Figure 2b,c). The limit of detection (LOD) was calculated to be 0.07 pg/mL in HEPES and 0.14 pg/mL in 10% diluted human serum.

DPV is a method used to express the measured current difference as a function of voltage by recording twice before the start of the pulse and immediately after the end of the pulse. This method can minimize the charging current and allow for a more accurate analysis of the Faraday current. Zhao group developed an electrochemical aptasensor based on a hybridization chain reaction and enzyme-signal amplification for interferon gamma (IFN-γ) measurement [48]. Figure 2d shows the schematic of an aptasensor. The recognition probe contained the sequence of an interferon gamma aptamer bound to the target substance. The unbound recognition probe binds to the capture probe on the electrode as an initiator to induce hybridization chain reaction. Using the recognition probe, the hairpin structures bio-H1 and bio-H2 were opened and bonded to the electrode. Next, streptavidin–alkaline phosphatase (SA–ALP) binds to biotin as a reporter molecule. SA–ALP converts 1-naphthyl phosphate, an electrically inactive material, into an electrically active derivative, 1-naphthol, to generate an electrochemical signal amplified by DPV. 1-Naphthol showed an oxidation peak near 230 mV in Tris-HCl buffer. The proposed sensor had an inverse relationship with IFN-γ concentration, which was quantified using DPV (Figure 2e,f) The sensor showed high sensitivity in the concentration range of 0.5–300 nM, with an LOD of 0.3 nM.

SWV applies a square wave voltage according to time and has the advantage of faster measurement speed than other voltammetry methods. The SWV pulse is composed of forward and reverse pulses, also called reduction and oxidation pulses, respectively. The current is plotted as the difference between the current values obtained from the oxidation and reduction pulses to obtain the voltammogram. Therefore, this method can minimize the current caused by capacitance. Recently, Lee group reported the fabrication of an aptasensor consisting of a multifunctional DNA three-way junction of an Au microgap electrode for IFN-γ detection [49]. The multifunctional DNA three-way junction was used as a bio-probe to detect IFN-γ with high sensitivity. The multifunctional DNA three-way junction was designed with a constant region (thiol group), an IFN-γ aptamer sequence, and a 4C-C (cytosine-cytosine) mismatch sequence (signal generation). Binding was achieved between the immobilized region of the multifunctional DNA three-way junction and the microgap electrode. An electrochemical signal was generated by intercalating four Ag ions into a mismatched sequence. The redox peak of the Ag ions was confirmed by CV. The signal intensity of the aptamer and the binding event according to the IFN-γ concentration was confirmed using SWV. A linear region was observed in the concentration range of 1 pg/mL to 10 ng/mL. The LOD was 0.67 pg/mL in a sample diluted in PBS (phosphate-buffered saline) and 0.42 pg/mL in a sample diluted in 10% human serum. This finding showed that this biosensor can be applied to both clinical and artificial samples.

### 2.3. Electrochemical Impedance Spectroscopy

EIS is a method for measuring the impedance of a measurement target for each frequency. The diameter of the semicircle observed at the high frequency of Nyquist in EIS represents the electron transfer resistance (R_ct_). Through measurement of R_ct_, it was possible to determine changes in the conductive properties of the working electrode. Mihaela group developed an impedimetric aptasensor for the label-free selective detection of interleukin-6 (IL-6) for colorectal cancer screening [50]. An aptasensor for high-sensitivity quantitative detection of IL-6 was fabricated using a glass-carbon electrode modified with p-aminobenzoic acid, p-aminothiolphenol, and gold nanoparticles. Figure 3a shows a schematic protocol and scheme of the aptasensor. A thiol-aptamer specific for IL-6 was immobilized on the modified electrode via a sulfur-gold bond. Next, using the aptamer as a bio-probe, label-free detection of the electrical properties that change according to the capture of the analyte was performed via EIS. In the Nyquist plot of EIS, the diameter of the semicircle represents the charge-transfer resistance (Rct). IL-6 concentration was quantified by changing the Rct value during the binding of the bio-probe and the analyte. IL-6 was thus measured in a physiological concentration range of 5 pg/mL to 100 ng/mL, and the LOD was calculated to be 1.6 pg/mL (Figure 3b).

Li group developed a label-free electrochemical impedance aptasensor based on target-induced exonuclease (Exo) inhibition to detect IFN-γ [51] (Figure 3c). IFN-γ was detected using a DNA hairpin with a loop of IFN-γ aptamer and a stem consisting of 5′-thiol modified as a probe. In the absence of IFN-γ, Exo digested single-stranded and double-stranded DNA hairpins, causing smaller impedance values on the electrode surface. In the presence of IFN-γ, the target and aptamer formed a complex, blocking the function of Exo, thereby preventing the DNA hairpin from being digested. In other words, impedance was larger on the electrode surface. The proposed aptasensor achieved a low detection limit of 0.7 pM over a wide range of 1 pM to 50 nM (Figure 3d) [51].

The Cristea group reported a label-free electrochemical aptasensor based on gold and polypyrrole nanoparticles for the detection of IL-6 [52]. To detect IL-6 in human serum, a highly sensitive electrochemical aptasensor was developed based on a carbon electrode modified with a nanocomposite composed of polypyrrole and gold nanoparticles. A favorable environment was provided for the immobilization of IL-6 aptamers through sulfur-gold bonds; that is, an environment simultaneously containing a conductive polymer and gold. Each manufacturing process of the aptasensor was monitored using CV and EIS. After completion of the optimization, IL-6 concentration was quantified using EIS. IL-6 content showed linearity in a wide concentration range of 1 pg/mL to 15 μg/mL, with an LOD of 0.33 pg/mL.

## 3. Optical-Based Detection

### 3.1. Optical Biosensor

Optical biosensors are analytical devices composed of an optical transducer system and bioreceptors, such as antigens, antibodies, nucleic acids, enzymes, cells, and tissues. They are characterized by high specificity, sensitivity, multi-sensing ability, and efficient cost. With these advantages, their use has grown exponentially over the past decade and has been widely used in various fields, such as biotechnology, environmental applications, and pharmaceuticals but there are limitations to the practical application of optical biosensors in academic and pharmaceutical environments. The main purpose of an optical biosensor is to generate a signal proportional to the concentration of the analyte to be measured [53,54]. Among various optical transducer systems, fluorescence, surface plasmon resonance (SPR), localized surface plasmon resonance (LSPR), and surface-enhanced Raman spectroscopy (SERS) are commonly used to detect cytokines (Figure 4).

### 3.2. Fluorescence

Fluorescence-based detection of cytokines is widely used owing to its high sensitivity. A fluorescent dye, such as a quantum dot, can visualize a corresponding fluorescent signal. The presence of an analyte can be confirmed by the change in the intensity of the fluorescent signal [55]. The advantages of this method include fast analysis time and high stability [56]. Tuleuova group detected IFN-γ using the quencher of fluorescence as a molecular marker that can measure fluorescence resonance energy transfer (FRET) by binding to the DNA strand that binds the aptamer. The detection limit is 5 nM, and the advantage is that the signal can be observed without the multiple washing steps required in standard immunoassays [57]. Zhang et al. designed two molecular beacons (MBs) and detected IFN-γ by amplifying the fluorescence signal using an MB track-mediated DNA walker and nicking enzyme. The detection limit was 7.65 fM, providing a sensitive platform for amplification analysis of various target molecules [58] (Figure 5a).

Liu et al. manufactured aptamers made of graphene quantum dots (GQDs) (Ap-GQDs) and epitope-modified GQDs (Ep-GQDs) for the detection of IFN-γ. The conjugate of Ap-GQD and Ep-GQD showed high sensitivity (2 pg mL^−1^), which showed its potential application for the detection of cell-secreted molecules [55]. Pan group inserted thiazole orange into a secondary G-quadruplex structure for IFN-γ detection and developed a strategy of reducing fluorescence by destroying the G-quadruplex structure and releasing thiazole orange when the target binds to the aptamer. The detection limit was 2 nM, and there were no complex modifications or chemical labeling, which provided simplicity and cost-effectiveness [64]. Wen group detected IFN-γ by making the first aptamer/protein/aptamer-polymer fluorescent sensor through DNA-based click polymerization. It had a detection limit of 1.63 fM and proved to be a convenient and highly selective sensor [56]. Kim et al. developed a fluorescent sensor that can detect IFN-γ using reduced graphene oxide nanosheets and had a detection limit of 0.1 ng/mL, providing a new rapid and specific detection method [59] (Figure 5b). Qin group amplified the fluorescence signal via microchip electrophoresis. The target material was IFN-γ, and the detection limit was 6.5 pM, which showed high sensitivity and specificity [65]. Wang et al. induced a fluorescence signal using a hairpin aptamer probe to detect IFN-γ. The detection limit was 0.6 pM, and this fluorescence signal amplification method provided efficient selectivity for the target [66]. Qin group formed a netlike hybridization chain reaction DNA nanostructure that provided effective signal enhancement for high-sensitivity fluorescence detection of IFN-γ. It had a detection limit of 1.2 pM, showing excellent selectivity and high sensitivity [67]. Qiu group manufactured a cell membrane-anchored sensor capable of detecting IFN-γ by combining aptamers and droplet microfluidics. The detection limit was 10 nM, and the sensor allowed for IFN-γ detection at the single-cell level [68]. Dhenadhayalan group detected IFN-γ using ReS_2_ and TiS_2_ nanosheet platforms. The detection limit was determined to be 57.6 pM for ReS_2_ and 82.7 pM for TiS_2_, revealing promising potential for future design of biosensors [60] (Figure 5c). Tuleuova et al. employed micropatterned poly(ethylene glycol) hydrogel microwells for IFN-γ detection, and then measured the fluorescence signal with avidin followed by biotin-aptamer fluorophore. The detection limit was 5 nM, indicating potential as a new strategy for IFN-γ detection [69]. Ma group designed an aptasensor for IFN-γ detection based on aggregation-induced emission, with a detection limit of 2 pg mL^−1^; this aptasensor provided a platform for monitoring IFN-γ secreted by cells [70]. Ghosh group developed an aptasensor for TNF-α detection based on FRET in an intracellular environment. The sensor was designed to enable cell-penetrating peptide-induced endocytosis, representing a novel measurement method [71].

### 3.3. SPR and LSPR

An SPR biosensor measures changes in refractive index using the resonance phenomenon of surface plasmon waves. Surface plasmon is a collective charge density oscillation of electrons occurring on the surface of a thin metal film, during which a surface plasmon wave is generated. Therefore, it is possible to observe the interaction between the bioprobe immobilized on the metal surface and the target material without using a label [72,73]. LSPR biosensor measurements are conducted based on metallic nanostructures with unique optical properties. Therefore, LSPR is different from SPR in that it vibrates locally in the nanostructure [74,75]. Chang group designed an aptamer hairpin structure to detect IFN-γ and produced an aptamer probe capable of binding to IFN-γ. The detection limit was 33 pM, providing the advantages of high sensitivity, reusability, and no additional labeling or sample preparation [61] (Figure 5d). Chuang et al. used a hairpin-shaped aptamer as a detection probe to specifically detect IFN-γ, and the aptamer was bound to streptavidin to amplify the signal; this aptamer had a detection limit of 10 pM and the experiment can be performed simply [76]. Berto group reported an electrolyte-gated organic field-effect transistor peptide aptasensor for TNF-α detection, which had a detection limit of 1 × 10^–12^ M. This is one of the first examples of organic electronic biosensors using a peptide aptamer as a key device for a gate electrode [77]. Bhalla group fabricated a nano-metal-insulator semiconductor sensor to detect IL-6 with a detection limit of 400 fM, which can be readily adopted for multiplexed and high-throughput label-free immunoassay systems, further driving innovations in biomedical and medical research [78]. Chuang group detected IFN-γ using aptamers and gold nanorods; they proposed a sensing method using an LSPR biosensing system and low-cost spectrometer [79]. Lin et al. utilized DNA-functionalized gold nanorods to increase the spectral shift of LSPR biosensing. The method can be used to detect IFN-γ (detection limit, 10 pM) and many other molecules [62] (Figure 5e).

### 3.4. SERS

Raman scattering is the process of scattering while losing or gaining energy. SERS is a biosensing technology based on the phenomenon in which Raman scattering intensity is rapidly increased when target materials are positioned between metal nanoparticles, and this sensing technology can detect various substances with a small number of samples, with high sensitivity [80,81]. Muhammad et al. fabricated a SERS biosensor composed of an aptamer and a gold nanoparticle array for IL-6 detection. Upon recognition of IL-6 in serum, the aptamer altered its structure, resulting in a corresponding change in the output Raman intensity ratio, thus allowing for quantitative evaluation. The detection limit was 0.8 pM, suggesting that the aptamer-based SERS biosensor is a promising tool for fast and convenient medical diagnostic applications [63] (Figure 5f).

## 4. Electrical Detection

### 4.1. Electrical Biosensor

Electrical biosensors can provide label-free detection of biomolecules, thus reducing the cost and time required for labeling processes, which are usually necessary in the application of optical biosensors. As the main components of electrical biosensors, field-effect transistors (FETs) and capacitors are integrated in modern microprocessors that process and store data. Because these electrical devices are massively fabricated with conventional semiconductor processes, electrical biosensors also have the potential for low-cost, massive fabrication with high reliability. The electrical biosensor detects a change in the electrical signal upon biomolecule interaction near the sensor surface (Figure 6). The dielectric medium of biomolecules filling the gap between the sensing electrode changes the capacitance as a sensing signal of a capacitive biosensor (Figure 6a). The electrical field or surface potential generated by the biomolecule modulates the FET current (Figure 6b). The use of aptamers as receptors is beneficial for electrical detection because the small size or conformation of aptamers near the sensor surface suppresses the effect of charge screening, which reduces the sensitivity of the electrical biosensors.

### 4.2. Capacitive Biosensor

A capacitive biosensor converts the binding event of a receptor and biomolecule between electrodes into a capacitance signal without a labeling process (Figure 6a). The capacitance can be described by the following equation [82]: $C=\varepsilon_{r}\varepsilon_{0}\frac{A}{d}$, where *A* is the surface area of the electrode, *d* is the distance between the electrodes, *ε_r_* is the dielectric constant of the medium between the electrodes, and *ε*_0_ is the permittivity of air (8.85 × 10^−12^ F/m). Capacitance can change owing to the specific binding of biomolecules between the electrodes, resulting in a change in the dielectric constant *ε_r_*. Liao group reported a capacitive biosensor that can detect platelet-derived growth factor BB (PDGF-BB), an important cytokine involved in neural inflammation [83]. They fabricated a biosensor based on a simple structure: an anti-PDGF-BB aptamer-modified silica wafer, where the 5′-phosphated aptamer was immobilized on an APTES-treated silicon wafer by EDC (Figure 7a).

The biosensor could detect a specific target (i.e., PDGF-BB) in the range of 50–1 μg/mL by monitoring the change in the capacitance, with a detection limit of approximately 40 nM (Figure 7b). Although the sensor surface was not passivated except the aptamer-modified area, the biosensor exhibited high specificity over control targets, such as bovine serum albumin, thrombin, and lysozyme. This result is attributed to a specific binding characteristic between aptamer and target molecule. Additional passivation process will further suppress a noise signal arising from non-specific binding between interfering molecules and sensor surface.

Several methods have been developed to improve the sensitivity of capacitive biosensors for cytokine detection. Kim group used an anodized aluminum oxide-based capacitive biosensor to detect IFN-γ, which is a biomarker of latent tuberculosis infection [84]. The nanoporous structure of the anodized aluminum oxide (AAO) membrane not only increased the surface area, leading to high sensitivity, but also inhibited nonspecific binding (Figure 7c). The biosensor could detect IFN-γ in human serum in the range of ~0.1 pg/mL to ~10 ng/mL, with an LOD of 0.2 pg/mL (Figure 7d). The characteristics of the biosensor were comparable with those of the commercial QuantiFERON-TB Gold ELISA kit, revealing the applicability of the biosensor as a diagnostic tool for latent tuberculosis infection. Although this sensor showed superior characteristics in the LOD compared to the ELISA counterpart, high variation in capacitance signal at higher concentration (>10^−10^ g/mL) makes it difficult to extract an accurate concentration (Figure 7d). Uniform immobilization of receptors on the nanostructure of the AAO membrane is needed for more reliable measurement of capacitance signals upon target binding. Qureshi group proposed a capacitive aptamer-based sandwich assay to detect vascular endothelial growth factor (VEGF)-165 in human serum [85]. Gold interdigitated microelectrodes were functionalized with anti-VEGF aptamers that first captured target VEGF proteins and then formed an aptamer-VEGF protein complex, followed by sandwiching with anti-VEGF antibody-conjugated magnetic beads (insets in Figure 7e,f). The capacitance signal was enhanced 3–8 times through this sandwiching method (Figure 7e,f). The capacitive aptamer–antibody-based sandwich assay detected VEGF protein in human serum in a dynamic range from 5 pg/mL to 1 ng/mL. The capacitance signal is proportional to the number of the magnetic beads. Aggregation of magnetic beads irrelevant to target binding can produce an error signal; thus, synthesis and storage of magnetic beads should be specially controlled and monitored to achieve uniform dispersion of magnetic beads. Chen group used a nanocomposite to improve the sensitivity of a capacitive biosensor to detect the inflammatory factor IL-3, a predictor of sepsis [86]. A complex longitudinal zeolite and iron oxide nanocomposite, which was modified on interdigitated microelectrodes, increased the immobilization of receptors on the surface to enhance the detection sensitivity, resulting in an LOD of 3 pg/mL in human serum. Although the sol–gel method in this work is a simple method for synthesis of the zeolite-iron nanomaterial, it is difficult to control size of grown nanomaterials, resulting in high size distribution. Optimal conditions in synthesis and deposition of the zeolite-iron nanomaterial may increase the sensor performance in terms of standard deviation of capacitance and LOD.

Capacitive biosensors have the advantages of high sensitivity, label-free detection, small size, and low cost, which enable point-of-care applications. Ceylan group reported a fast, low-cost, hand-held point-of-care diagnostic device based on interdigitated capacitive biosensors to detect multiple biomarkers, including TNF-α and IL-6 [87]. A 2 × 6 interdigitated capacitive array was fabricated on a disposable cartridge in a circular shape, providing sample droplet impingement and better contact. A change in the capacitance upon target binding was identified by a capacitance-to-digital converter integrated circuit. The shelf-life of the reactivated ready-to-use cartridges was 3 months under optimal conditions. The biosensor could analyze six different biomarkers in real-patient blood samples within less than 30 min. Despite of multiplexed detection of biomarkers and long self-life, selectivity of the sensor between different biomarkers should be confirmed to improve the diagnostic accuracy by compensating the noise signal arising from non-specific binding.

### 4.3. Field-Effect Transistor Biosensor

An FET is a switching or amplifying device in which electrical current flowing through a semiconductor channel between two metal electrodes (i.e., the source and the drain) is controlled by the voltage applied to an additional electrode (i.e., the gate) [88]. In an FET biosensor (Figure 6b), a solid-state gate is replaced with receptors, such as an aptamer, antibody, and enzyme, to capture target biomolecules that act as a “bio-gate” to modulate the electrical current according to the concentration of analytes [89]. The advantages of aptamer-functionalized FET biosensors include high sensitivity, fast analysis, and high portability, which can be realized through the label-free electrical detection, the small size of FETs, and their high compatibility with electrical readout circuits. The conformational change of the negatively charged aptamer upon target binding can modulate the surface potential near the semiconducting channel, resulting in a change in the channel current [90].

Wang group showed an aptamer-functionalized monolayer graphene as a conducting channel of FET to detect TNF-α, an inflammatory cytokine biomarker (Figure 8a) [91].

In the first step of graphene functionalization, 1-pyrenebutanoic acid succinimidyl ester (PASE), was immobilized on a monolayer graphene through π–π stacking as a linker for aptamer functionalization [94]. Next, the 5′-phosphated aptamer was covalently bonded to the PASE molecule, resulting in aptamer immobilization on the graphene surface. Finally, the graphene was treated with Tween 20 and ethanolamine to passivate the uncoated area of graphene and quench the unreacted PASE molecules. As the TNF-α concentration increased, the Dirac point voltage (*V_Dirac_*) decreased, indicating n-type doping caused by the specific binding between the aptamer and TNF-α (Figure 8b). The LOD for TNF-α was 5 pM. The biosensor showed high selectivity for two other inflammatory cytokines, namely IFN-γ and IL-002, as well as bovine serum albumin (Figure 8c). Based on good characteristics of low LOD and high selectivity, this sensor has the potential to be applied in a serum sample. Further analysis is needed to examine the sensing mechanism such as induced charge from TNF-α to graphene or binding-induced conformational change of aptamer. The latter is suitable for overcoming double-layer screening, which is dominant in physiological fluids (i.e., a serum sample).

Several efforts have been made to enhance the sensitivity and selectivity of aptamer-functionalized FET biosensors for the detection of cytokine biomarkers. Hwang group reported a grumbled graphene FET biosensor for the detection of IL-6 protein with aM-level sensitivity (Figure 8d,e) [92]. The extremely low LOD is caused by the two effects of the bending of graphene: (1) increased Debye screening length, which reduced the charge screening of the biomolecules, and (2) bandgap opening, which allowed for an exponential response in current from a small number of charges [95]. Because the sensitivity is controlled by the crumpling ratio, a wide range of target concentrations can be covered by preparing several sensors with different crumpling ratios. The uniform control of the crumpling ratio can improve the reliability in biosensing. Hao group showed the modulation of the density of PASE molecules that immobilize an aptamer by tuning the electric field to improve the detection sensitivity (Figure 8f) [93]. Application of −0.3 V electric field for 3 h during the PASE immobilization process increased the PASE and aptamer immobilization densities, as confirmed by electrical characterization to measure a shift in the Dirac point voltage (*V_Dirac_*) (Figure 8g) and EDS characterization to quantify the phosphorus observed in the aptamer (Figure 8h). With the electric field method, the LOD for IL-6 biomarker was reduced from 1.66 pM to 618 fM. This work sheds light on the importance of linker density on sensing performance. Despite the novel method to improve the LOD, the Ag/AgCl electrode inserted in the PASE solution has a bulk size to prevent miniaturization. A further approach is required to achieve a portable system by minimizing the sensor component (i.e., Ag/AgCl electrode) for the electric field generation. Wang group proposed a novel method to measure cytokine levels in undiluted sweat, where high background interference (e.g., lactic acid and amino acids) exists [95]. They fabricated a graphene-Nafion composite film by drop-casting a Nafion solution on a graphene surface. A porous and negatively charged Nafion film screened out interfering molecules. In addition, the graphene-Nafion composite film provided regenerative capability to the biosensor up to 80 cycles by removing the Nafion film. The biosensor was capable of detecting IFN-γ in undiluted human sweat, with a range from 0.015 to 250 nM and an LOD as low as 740 fM. The specificity of the biosensor in targeting a target biomarker (i.e., IFN-γ) was confirmed using control biomarkers, such as TNF-α, IL-2, and IL-6, which are inflammatory cytokines closely related to IFN-γ. The thickness of the Nafion film was ~50 nm, indicating that the distance between target biomolecule and graphene is larger than 50 nm. Because the high signal occurs when the target biomolecule is close to the channel, the effect of the Nafion thickness on the sensitivity could be the subject of future research.

## 5. Strategies for Improving Sensor Performance

### 5.1. Nanomaterials

Despite the advantages of aptamers such as low cost, ease of fabrication, and high stability, avatar sensors have limited application due to low specificity and sensitivity [27]. Several methodologies are presented in this section to overcome these shortcomings and gain a competitive edge, for example, by directly improving the sensitivity of a sensor, enabling reuse for cost advantage, or adding other features.

Nanotechnology is a technology for synthesizing and controlling the assembly of nanometer-sized materials, and it has advanced considerably over the past few decades [96,97,98]. Because of their unique properties, nanomaterials are applied in various fields [99,100,101]. In particular, with respect to biosensors, various nanoparticles can be used to improve the efficiency and sensitivity of both transducers and receptors [102]. This is because of the high area-to-volume ratio [49], optical properties (such as luminescence, fluorescence, refractive index, light scattering [103]), and high electrical activity [104] of nanoparticles.

Ghalehno group improved the performance of a sensor by introducing gold nanoparticles to a traditional sandwich ELISA sensor [105]. A TNF aptamer was immobilized on a graphite electrode modified with cobalt hexacyanoferrate and gold nanoparticles, and an antibody conjugated to horseradish peroxidase was used as a secondary antibody. The fabricated sensor had a detection range of 1–100 pg/mL and a detection limit of 0.52 pg/mL.

In addition, in the study by Cristea et al., gold nanoparticles were used to improve the active surface area of an electrode, thereby suppressing fouling, which can occur in hard and flat electrodes, and improving the analytical performance [52] (Figure 9a). The number of aptamer molecules immobilized on the electrode surface was increased by immobilizing gold nanoparticles on the electrode surface and fixing the aptamer onto the gold nanoparticles. The above method can effectively increase the amount of aptamer immobilized on the surface and substantially enhance the electrical activity; thus, it can be widely used in electrochemical sensors.

Another study also reported label-free detection of IL-6 using the refractive index change in nanomaterials [106] (Figure 9b). LSPR is attractive because it enables fast, label-free, real-time monitoring of biomolecular binding events [107]. According to Tamiya group, the nanoimprint method was used to fabricate nanopillar structures and to detect IL-6 release from cells using microwell arrays. In another study, Chen and colleagues simultaneously measured the cytokines IL-6, TNF-α, IL-10, and IL-4 without labeling in adipose tissue chips by fixing antibodies to gold nanorods [108], with a short detection time of less than 30 min and an excellent detection limit of 20 pg/mL.

### 5.2. Microfluidic System

Biosensors are useful for quantifying analytes in samples; however, measuring dynamic changes in analyte concentrations is highly difficult. To address this problem, microfluidic devices have been utilized in studies on biosensors [109,110,111]. A microfluidic device processes and screens fluids of varying volume ranging from nanoliters to microliters. The combination of a microfluidic device and a detection system enables real-time analysis of a small sample size. In addition, because the fluid flow can be controlled by a laminar flow, highly reproducible analysis is possible [112]. Moreover, an understanding of cytokine secretion by single cells is essential for developing novel therapies for multiple diseases [113].

A study by Altug group provided an excellent example of a combination of microfluidic devices and sensing systems [114]. In this study, the authors used a microfluidic device that can accurately monitor cytokine secretion in single cells in real time, coupled with a label-free nanoplasmonic biosensor (Figure 10). A uniform nanohole array was fabricated on the entire sensor surface through photolithography. Cytokines were detected by the fabricated nanohole structure through extraordinary optical transmission, as this phenomenon responds sensitively to changes in the refractive index of the nanohole surface; thus, when molecular bonding occurred on the sensor surface, the spectrum of the peak wavelength shifted. This enabled real-time monitoring of molecular binding events (Figure 10d). Moreover, the authors designed a novel microfluidic device because existing continuous-flow microfluidic devices inhibit the accumulation of cytostatics, making them impractical for single-cell analysis. A pneumatically actuated microvalve was used to separate the incubation chamber and the fluid (Figure 10b). The lower layer contained the cell culture, and the upper layer contained the pneumatic lines and actuation chambers required to operate the microvalve. A tortuous hydraulic channel was designed to control the humidity and temperature in the chamber. IL-2 was detected using the manufactured device, with a high detection limit of 39 pg/mL. This method allowed for continuous monitoring of cytokines in EL4 lymphoma cells for hours without interrupting the cell culture.

Another example is a study measuring the release level and release time of IL-6 and TNF-α using a 3D MuscleChip [115]. In this study, a 3D MuscleChip was fabricated on an interdigitated electrode using an in vitro biomimetic tissue model, and it was able to confirm the release of IL-6 and TNF-α from skeletal muscle tissues induced by electrical and LPS (Lipopolysaccharide) stimulations. This 3D MuscleChip was intended to be used for understanding and treating muscle metabolic disorders. The author succeeded in culturing highly aligned muscle tissues on the chip using GelMA-CMCMA (Gelatin methacryloyl- Carboxymethyl cellulose methacrylate) hydrogel, and a microfluidic network was applied to promote cytokine secretion by the muscle tissue fixed on the chip. During the relaxation time, IL-6 and TNF-α were detected at 1 μg/mL and 10 ng/mL, respectively. The maximum detection peak was reached at 1 h after LPS stimulation, and IL-6 was detected at 2.5 μg/mL. The concentration of released TNF-α increased slowly and continuously. This study revealed the release of various cytokines induced by electrical and biological stimuli, providing a better understanding of muscle growth processes and inflammatory responses.

### 5.3. Reusable Biosensor

Biosensors can rapidly detect biomarkers, such as pathogens or proteins, in vitro. In addition, as they can be developed into portable diagnostic devices, efforts are being made to advance their point-of-care applications [116]. However, one of the biggest barriers to the commercialization of biosensors is the manufacturing cost. To reduce this cost, research on the application of aptamers [117], the development of inexpensive electrode materials [118], and the utilization of multiple analytical systems [119] is in progress. Here, we present a reusable biosensor as a new approach for reducing the manufacturing cost of biosensors.

Liu group fabricated a molecularly imprinted polymer (MIP)-based biosensor for IL-1β detection that can be reused more than three times with a coefficient of variation of 2.08% [120] (Figure 11a).

The stainless-steel surface was first transformed into a polydopamine layer, and then a polyethyleneimine layer was attached by electro-adsorption. The target molecule was then adsorbed onto the polyethyleneimine-terminated stainless-steel surface. Finally, the MIP biosensing device was manufactured by removing the target molecule and leaving a hole of a specific shape. IL-1β was detected based on fluorescence intensity by incubating IL-1β with a fluorescently tagged IL-1β detection antibody. The manufactured sensor had a low detection limit of 10.2 pg/mL and a detection range of 25–400 pg/mL. To reuse it, the MIP sensing interface was washed with a mixture of methanol and hydrochloric acid and then washed with a mixture of PBS (phosphate-buffered saline) and Tween-20. The fluorescence intensity of the MIP biosensing device changed slightly after the three regeneration cycles. The coefficients of variation after three and five uses were 2.08% and 7.48%, respectively, allowing reuse.

In another study, a biosensor was generated for SWV-based IFN-γ detection using the aptamer developed by Revzin et al. [121] (Figure 11b). The IFN-γ aptamers were modified with thiol groups, immobilized on gold electrodes, and bound to methylene blue redox labels. When the binding event of IFN-γ and IFN-γ occurred, the hairpin structure of the aptamer was modified, causing methylene blue to move away from the electrode, and thus reducing the efficiency of electron transfer. IFN-γ in the sample was detected through the change in current according to the change in electron transfer efficiency.

The change in redox current was confirmed by SWV, and the detection limit of the manufactured biosensor was 0.06 nM. Aptamer reuse in the manufactured sensor was possible owing to the high chemical stability of the aptamer. The IFN-γ aptamer and IFN-γ complex was reused after the complex was destroyed by treatment with urea buffer. The experiment confirmed that the aptasensorcould be reused up to 10 times.

### 5.4. Wearable Biosensor

Wearable biosensors are gaining attention with the advancement of point-of-care diagnostics because they can receive biosignals from the wearer in real time [122,123]. Human biofluids that can be obtained noninvasively, such as saliva, tears, and sweat, are preferable over fluids that must be collected invasively, as they are associated with less pain and no risk of infection [124]. Additionally, these human biofluids contain many cytokines, making them attractive samples [125]. Moreover, because a wearable sensor must be attached to the surface of the human body, the substrate of the sensor must be flexible, and the signal must be constant despite deformation [126].

Zhao have constructed such a wearable sensor [124] (Figure 12a). A GFET using graphene as a channel was fabricated on a 2.5-μm ultra-thin polymer substrate, and Tween 80 was used to suppress nonspecific adsorption to the graphene surface. Aptamers of TNF-α and IFN-γ, which are representative cytokines, were used in the manufactured sensor for high-sensitivity detection, resulting in detection limits of 2.75 pM and 2.89 pM, respectively. The study also showed that real-time monitoring of cytokines in artificial tear samples is possible even with sensor modification such as stretching or banding, resulting in a consistent sensing response.

In another example, Pan et al. fabricated a reusable wearable sensor using a graphene-Nafion film [127] (Figure 12b). Channels composed of graphene-Nafion composite films enhanced biosensor regeneration while minimizing nonspecific adsorption. The fabricated biosensor can sensitively detect cytokines in undiluted human sweat with a detection range of 0.015–250 nM and a detection limit of 740 fM. It was also capable of up to 80 replays, and showed a consistent detection response in as many as 100 crumpling tests.

## 6. Conclusions

There is a demand for sensors capable of detecting cytokines. In particular, cytokine detection using aptamers is expected to be commercialized owing to the favorable characteristics of aptamers. In this review, we have summarized the latest advances on the application of aptasensors in various methods for cytokine detection (Table 1). Various methods for improving the effectiveness of aptasensors are also discussed. For example, nanomaterials can be used to substantially improve the sensitivity of sensors. Moreover, microfluidic systems enable the real-time detection of cytokines. Despite the development of various cytokine detection sensors, ELISA is still the most widely used cytokine detection tool because of its multiple detection capabilities. Considering that cytokines influence each other, simultaneous monitoring of multiple cytokines is necessary [2]. For this purpose, nanoarrays and multiple sensing technologies have been devised, but they are still in their infancy. Studies on the detection of cytokines using aptamers are few despite the advantages of aptamers as biosensors (such as low production and transportation costs). Nevertheless, aptamers show that cytokine detection is possible with high sensitivity (Table 1). Therefore, research on cytokine detection using aptamers has high growth potential. In addition, investment and research on cytokine detection will continue to increase. Interest in cytokines is further increasing due to the impact of COVID-19 [128].

## Figures and Tables

**Figure 1 sensors-21-08491-f001:**
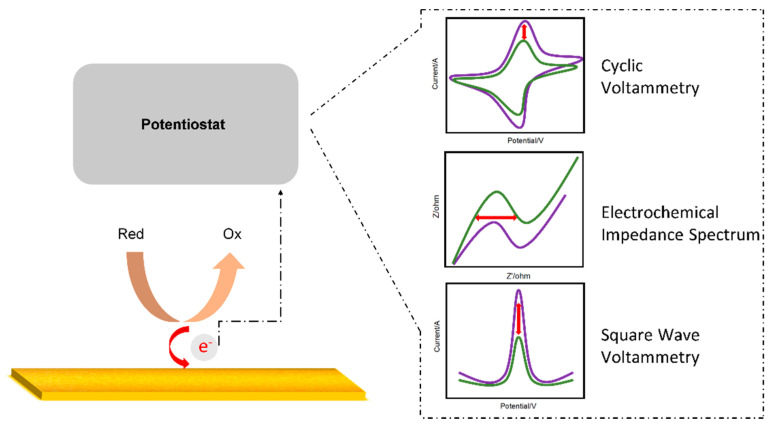
Schematic diagram of an electrochemical sensor.

**Figure 2 sensors-21-08491-f002:**
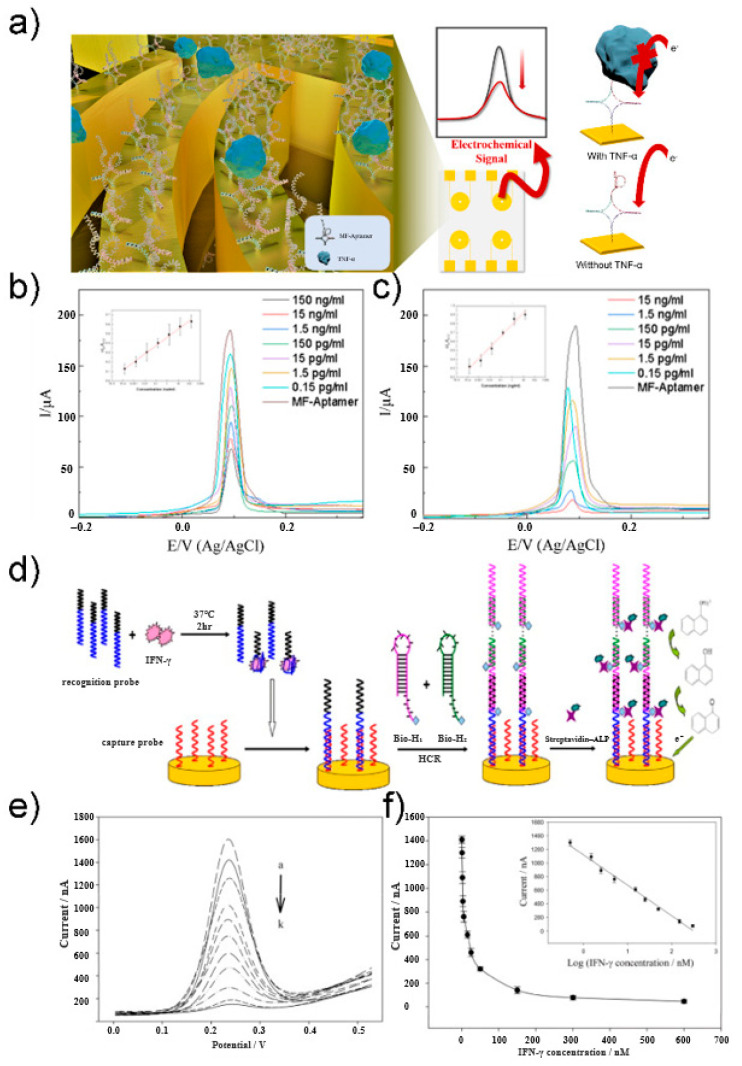
(**a**) Schematic illustration of TNF-α electrochemical biosensor. (**b**) CV at different concentration of TNF-α diluted with PBS buffer in 10 mM HEPES and 5 mM [Fe(CN)_6_]^3−/4−^. (**c**) CV at different concentration of TNF-α diluted with 10% human serum buffer in 10 mM HEPES and 5 mM [Fe(CN)_6_]^3−/4−^. Reprinted with permission from [47]. Copyright 2021 Elsevier. (**d**) Schematic illustration of IFN-γ electrochemical aptasensor. (**e**) DPV at different concentration of IFN-γ. (**f**) Linear regression curve of different IFN-γ concentrations. Reprinted with permission from [48]. Copyright 2012 Elsevier.

**Figure 3 sensors-21-08491-f003:**
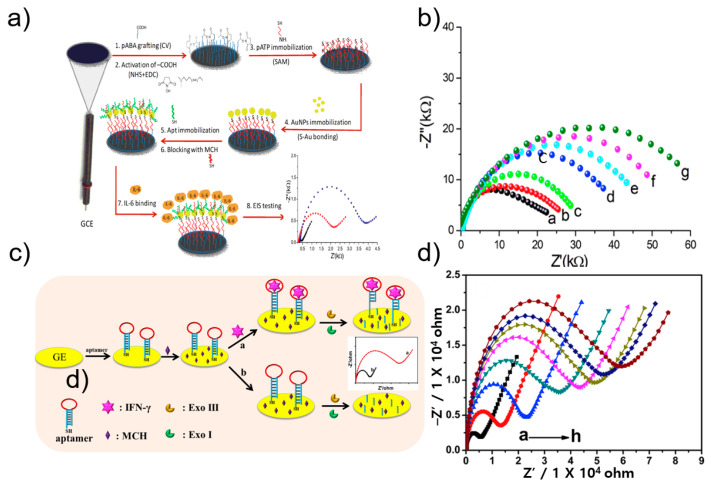
(**a**) Schematic illustration of IL-6 electrochemical aptasensor. (**b**) EIS at different concentration of IL-6. Reprinted with permission from [50]. Copyright 2019 Elsevier. (**c**) Schematic illustration of IFN-γ impedance electrochemical aptasensor. (**d**) EIS at different concentration of IFN-γ. Reprinted with permission from [51]. Copyright 2019 Elsevier.

**Figure 4 sensors-21-08491-f004:**
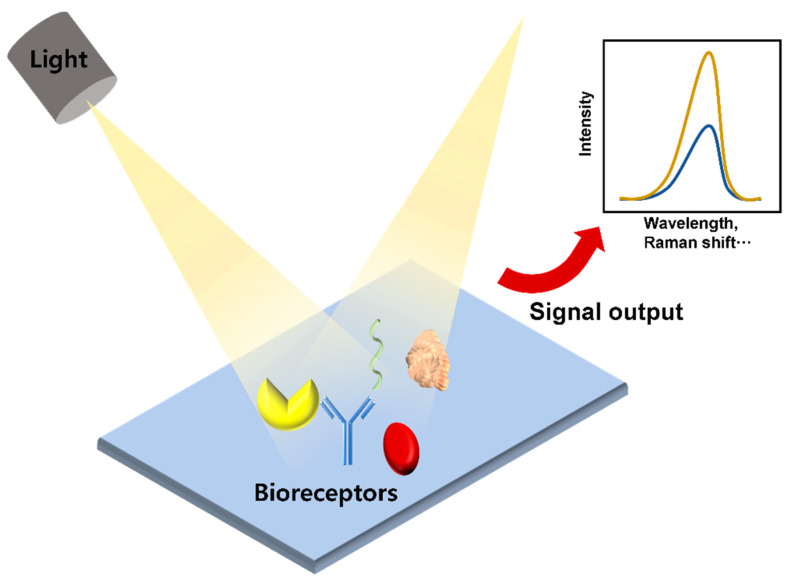
Schematic diagram of an optical sensor.

**Figure 5 sensors-21-08491-f005:**
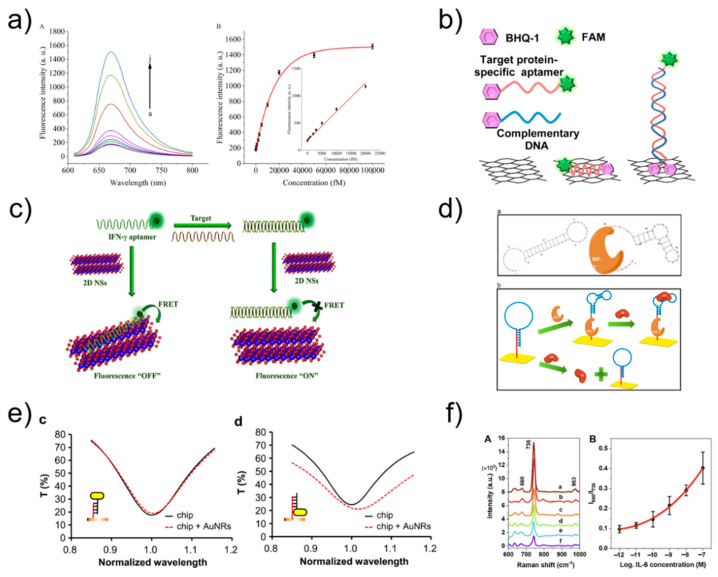
(**a**) Fluorescence emission spectra at different IFN-γ concentration and relationship between the fluorescence intensity and the concentration of IFN-γ. Reprinted with permission from [58]. Copyright 2018 Elsevier. (**b**) Schematic illustration of ssAptamers and dsAptamers anchored on a rGO nanosheet. Reprinted with permission from [59]. Copyright 2014 Elsevier. (**c**) fluorescence sensing mechanism of IFN-γ. Reprinted with permission from [60]. Copyright 2018 Elsevier. (**d**) Schematic representation of IFN-γ optical aptasensor. Reprinted with permission from [61]. Copyright 2012 Elsevier. (**e**) Transmission spectra of the LSPR chip before (black solid line) and after (red dashed line) the AuNRs were immobilized on the chip surface in the way shown in the insets. Reprinted with permission from [62]. Copyright 2016 Elsevier. (**f**) SERS spectra using aptamer-modified Au NPs array substrate and Corresponding SERS intensity ratios (I_660_/I_736_) versus IL-6 concentrations for obtaining the standard curve. Reprinted with permission from [63]. Copyright 2021 Elsevier.

**Figure 6 sensors-21-08491-f006:**
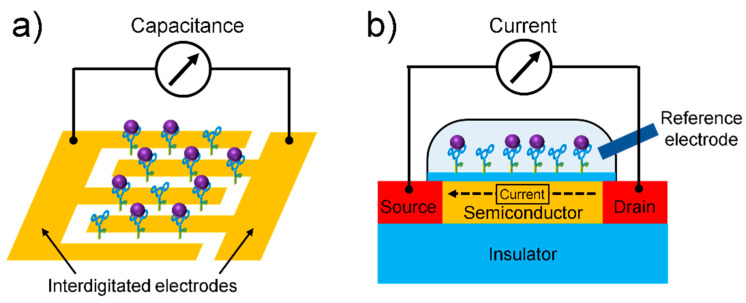
Schematics of (**a**) capacitive biosensor and (**b**) FET-based biosensor.

**Figure 7 sensors-21-08491-f007:**
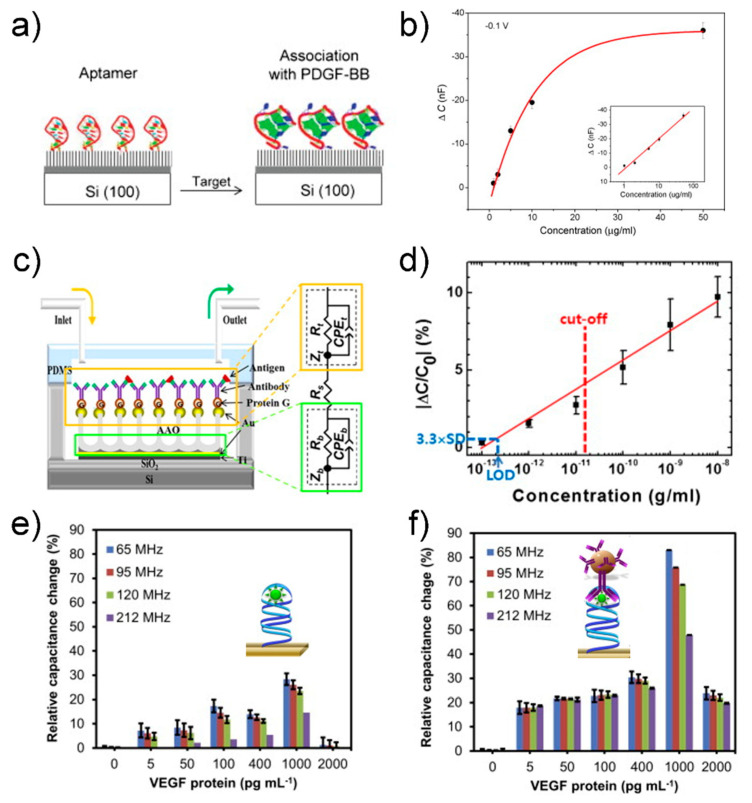
(**a**) Schematic of aptamer immobilization and protein association: (**left**) aptamer-immobilized surface through phosphate-amino covalent linkage and (**right**) surface after incubation with PDGF-BB. (**b**) Concentration profile for impedance sensing of protein–aptamer interactions. The anti-PDGF-BB aptamer-modified silica wafers were incubated with increasing concentrations of PDGF-BB: 1 μg/mL, 2 μg/mL, 5 μg/mL, 10 μg/mL and 50 μg/mL. The calibration plots in both the linear and logarithmic scale (inset) are presented. Reprinted with permission from [83]. Copyright 2007 Elsevier. (**c**) Schematic diagram of the AAO-based capacitive sensor. Each biomolecule is assembled from bottom to top. (**d**) Calibration curve measured with the AAO-based capacitive sensor. The red dashed line indicates the cut-off concentration at 15 pg/mL of IFN-γ for LTBI and LOD is the limit of detection estimated by assuming 3.3 × SD ≈ 0.46%. Reprinted with permission from [84]. Copyright 2014 Elsevier. Relative percent changes in capacitance responses occurred at different frequencies (65, 95, 120 and 212 MHz) with; (**e**) primary complex (aptasensor, before sandwiching) and (**f**) secondary complex (apta-immunosensor, after sandwiching with MB-Abs) formed on the sensor surfaces. The insets in figures (**e**,**f**) represents illustrations before and after sandwiching of aptamer–VEGF protein complex with MB-Abs. Reprinted with permission from [85]. Copyright 2015 Elsevier.

**Figure 8 sensors-21-08491-f008:**
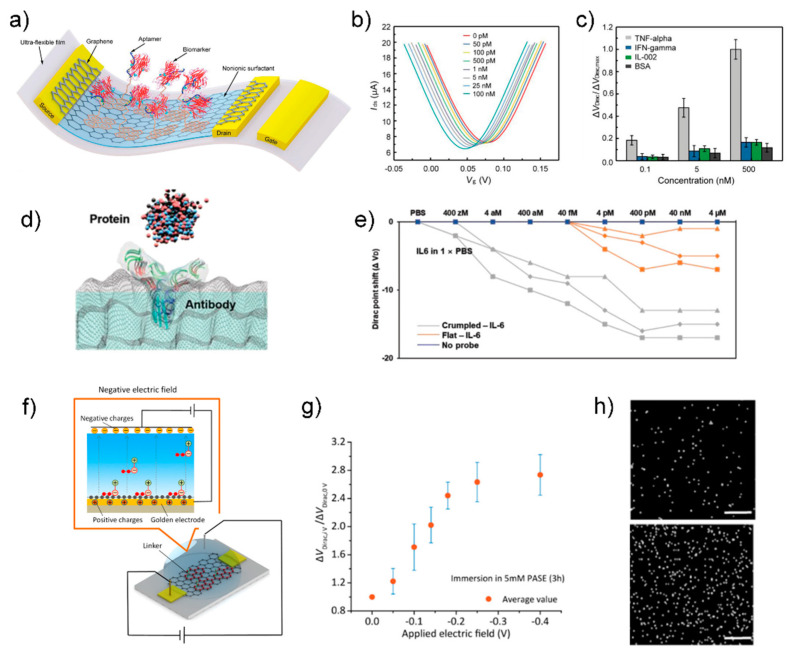
(**a**) Schematic of the aptamer-functionalized graphene FET biosensor. (**b**) Transfer characteristic curves measured when the biosensor was exposed to TNF-α solution with different concentrations. (**c**) The normalized Dirac point shift Δ*V_Dirac_*/Δ*V_Dirac,max_* showing the response of the biosensor to different concentrations (0.1, 5, and 500 × 10^−9^ M) of TNF-α and the control proteins (IFN-γ, IL-002, and BSA). Reprinted with permission from [91]. Copyright 2019 WILEY-VCH GmbH. (**d**) Schematic of proteins capturing with specific antibodies on the crumpled graphene channel. (**e**) Dirac voltage shift of the FET sensor with detection of IL-6 protein. Reprinted with permission from [92]. Copyright 2021 WILEY-VCH GmbH. (**f**) Diagram of PASE immobilization with applying negative electric field. With applying negative electric field through the inserted Ag/AgCl electrode, PASE molecules would be arranged regularly with directivity with pyrenyl groups forced toward the graphene surface due to the electrostatic repulsion, making further quantities of PASE molecules anchored on the graphene through π–π stacking and hence increasing the PASE immobilization density. (**g**) Dirac point shift Δ*V_Dirac_*/Δ*V_Dirac-0_* is plotted as a function of the applying electric field voltage. Here, Δ*V_Dirac-0_* and Δ*V_Dirac_* are measured after graphene immersion in 5 mM PASE at ~25 °C for 3 h without and with applying negative electric field at a given voltage value. (**h**) EDS characterization results of graphene surfaces without (**top**) and with (**bottom**) applying electric field during the PASE and aptamer immobilization process. White dots represent the parts covered with phosphorus, which is a main constituent element of aptamer and not contained in PASE. Scale bar: 1 μm. Reprinted with permission from [93]. Copyright 2020 American Chemical Society.

**Figure 9 sensors-21-08491-f009:**
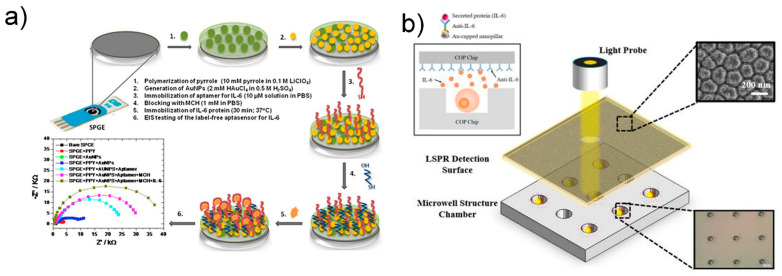
(**a**) Schematic diagram of label-free electrochemical biosensor. Reprinted with permission from [52]. Copyright 2017 Elsevier. (**b**) Schematic of the LSPR cdevice for sytokine detection. Reprinted with permission from [106]. Creative Commons Attribution License.

**Figure 10 sensors-21-08491-f010:**
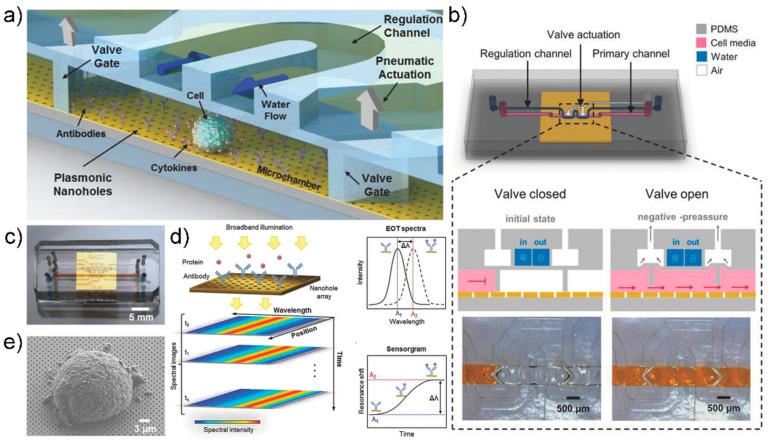
(**a**) Schematic illustration of the optofluidic platform for single cell-analysis (**b**) Schematic representation of microfluidic system design and valve actuation mechanism. (**c**) photo image of the integrated device (**d**) Appearance of EOT resonance peak due to nanohole array. Depending on the binding of the analyte, the wavelength of the resonance peak changes due to the change in the refractive index of the plasmon surface. (**e**) SEM image of a EL4 Cell on the nanohole array. Reprinted with permission from [114]. Copyright 2018 WILEY-VCH GmbH.

**Figure 11 sensors-21-08491-f011:**
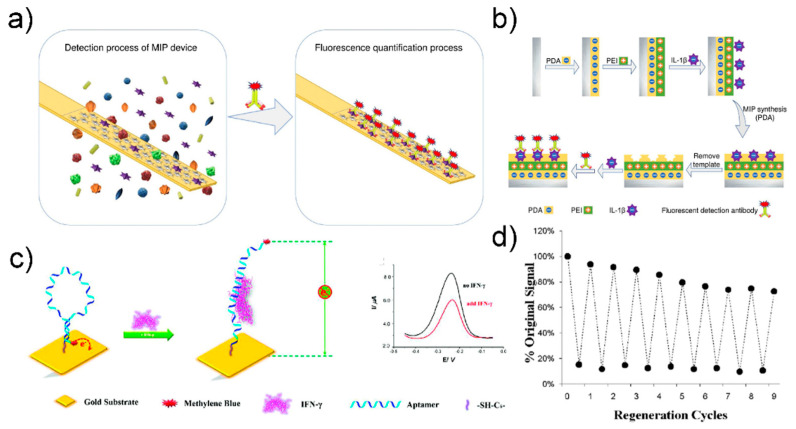
(**a**) Schematic image of the MIP based biosensor for cytokine detection (**b**) fabrication step of MIP based biosensor. Reprinted with permission from [120]. Copyright 2019 Elsevier. (**c**) Schematic image of SWV based IFN- γ detection biosensor (**d**) Evaluation of the regeneration of the sensor through urea washing. Reprinted with permission from [121]. Copyright 2010 American Chemical Society.

**Figure 12 sensors-21-08491-f012:**
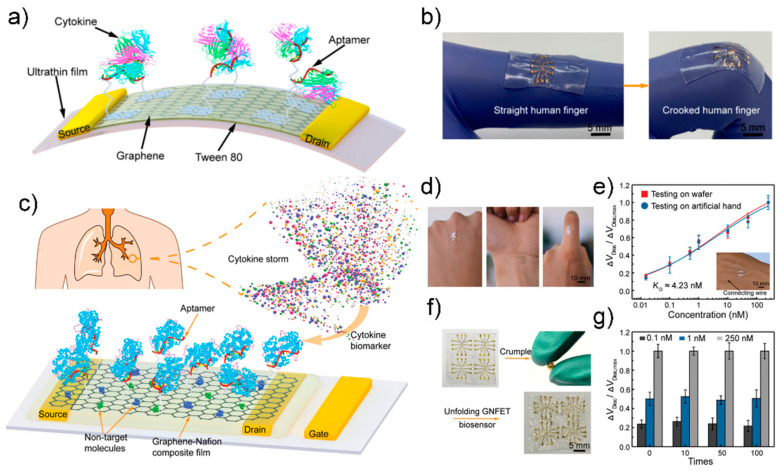
(**a**) Schematic diagram of flexible GFET biosensor (**b**) Fabricated biosensor can be stretched with the human body. Reprinted with permission from [123]. Creative Commons Attribution License. (**c**) Schematic diagram of the graphene-nafion FET biosensor (**d**) Photograph of the fabricated biosensor mounted on the human body (**e**) Normalized Dirac point shift in undiluted human sweat (**f**) Photograph of the crumping process (**g**) Sensing response of the fabricated biosensor various cytokine concentration at crumpling cycles. Reprinted with permission from [127]. Copyright 2018 WILEY-VCH GmbH.

**Table 1 sensors-21-08491-t001:** Summary of studies cited in this review.

Target Molecule	Detection Method	Detection Range	Limit of Detection	Reference
TNF-α ^a^	CV	0.15 pg/mL–150 g/mL	0.14 pg/mL	[47]
IFN-γ ^b^	DPV	0.5–300 nM	0.3 nM	[48]
IFN-γ	SWV	1 pg/mL–10 ng/mL	0.42 pg/mL	[49]
IL-6 ^c^	EIS	5 pg/mL–100 ng/mL	1.6 pg/mL	[50]
IFN-γ	EIS	1 pM–50 nM	0.7 pM	[51]
IL-6	EIS	1 pg/mL–15 μg/mL	0.33 pg/mL	[52]
IFN-γ	Fluorescence	5–100 pg/mL	2 pg/mL	[55]
IFN-γ	Fluorescence	0.01 pM–10 nM	1.63 fM	[56]
IFN-γ	Fluorescence	5–100 nM	5 nM	[57]
IFN-γ	Fluorescence	0–20 fM	7.65 fM	[58]
IFN-γ	Fluorescence	0.1 ng/mL–10 μg/mL	0.1 ng/mL	[59]
IFN-γ	Fluorescence	0–400 pM (ReS_2_)	57.6 pM (ReS_2_)	[60]
IFN-γ	Fluorescence	0–300 pM (TiS_2_)	82.7 pM (TiS_2_)	[60]
IFN-γ	SPR	0.3–333 nM	33 pM	[61]
IFN-γ	LSPR	0.01–1 nM	10 pM	[62]
IL-6	SERS	10^−12^–10^−7^ M	0.8 pM	[63]
IFN-γ	Fluorescence	3–120 nM	2 nM	[64]
IFN-γ	Fluorescence	15 pM–2.5 nM	6.5 pM	[65]
IFN-γ	Fluorescence	0.001–50 nM	0.6 pM	[66]
IFN-γ	Fluorescence	5–1000 pM	1.2 pM	[67]
IFN-γ	Fluorescence	17.2 nM–550 nM	10 nM	[68]
IFN-γ	Fluorescence	5–100 nM	5 nM	[69]
IFN-γ	Fluorescence	0–100 pg/mL	2 pg/mL	[70]
TNF-α	Fluorescence	0.34–17 nM	0.34 nM	[71]
IFN-γ	SPR	0.01–100 nM	10 pM	[76]
TNF-α	SPR	1–10 pg/mL	1 × 10^−12^ M	[77]
IL-6	LSPR	1 pM–100 nM	400 fM	[78]
IFN-γ	LSPR	0.1–10 nM	0.1 nM	[79]
PDGF-BB ^d^	Capacitance	1–50 μg/mL	~1 μg/mL	[83]
IFN-γ	Capacitance	0.1 pg/mL–10 ng/mL	0.2 pg/mL	[84]
VEGF ^e^	Capacitance	5 pg/mL–1 ng/mL	401 pg/mL	[85]
IL-6	Capacitance	2.5–20 ng/mL	2.5 ng/mL	[87]
TNF-α	Capacitance	0.3–50 ng/mL	0.3 ng/mL	[87]
TNF-α	FET	100 nM–50 pM	5 pM	[91]
IL-6	FET	50 nM–1 pM	618 fM	[93]
IFN-γ	FET	250 nM–15 pM	740 fM	[127]

^a^ Tumor necrosis factor-alpha: ^b^ Interferon-gamma; ^c^ Interlukin-6; ^d^ Platelet-derived growth factor BB; ^e^ Vascular endothelial growth factor.

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
