# Peer review of "Recent Advances in Aptasensor for Cytokine Detection: A Review"

_sensors, 2021, doi:10.3390/s21248491_

Round 1
Reviewer 1 Report
Dear Editor,
The manuscript entitled “Recent advances in aptasensor for cytokine detection: A review” by Kim et al. describes applications of aptamers-based biosensors for cytokine detection. Various detection methods are described including electrochemical, optical, and electrical readouts Also, several approaches for detection signals enhancement are discussed.
In my opinion, the manuscripts’ objective and perspective are very interesting, and the manuscript should be accepted for publication after minor revisions. The manuscript is well-written in general and a positive aspect is that the authors included many of the original figures of the discussed manuscripts.
My comments for the authors to consider are:
- The abstract should be re-written: the authors should mention briefly what aptamers are, cytokines are the ‘heart’ of the manuscript but they are not mentioned adequately in the abstract and the manuscript sections brief describe is missing.
- Please explain what ‘nd’ stands for in page 1, line 32.
- The paragraph about aptamers (page 2, lines 56-69) should be re-written; it is very confusing as it is.
- In page 2, lines 95-98 two sentences are repeating the same thing. Please re-phrase them.
- The paragraph in page 3, lines 103-116 contains too much details which are repeated in sections 2.2 and 2.3, therefore it could be minimized.
- Reference 50 is used twice for 2 different works, page 6, lines 176-187 and page 7 lines 198-207. Please correct.
- In page 13, line 360 a reference is missing.
- When the authors use the phrase ‘author et al’ they should put the first author. If they want to use the last authors’ name, they should say ‘last authors’ group’. It is confusing to find the reference the way it is.
- Please make a table containing all mentioned aptasensors, the analyzed cytokine, their linear range and LOD to facilitate comparison of the presented methods.
Overall, it is an interesting manuscript and should be accepted after minor revisions, in my opinion.
Reviewer 2 Report
Overall, the review paper does not offer useful insights, it just summarizes previous work in a positive way, ignoring the main problem(s) with apasensors.
Only very little discussion is provided, and more discussion should be added on the aspects of limitations, translation from lab to market, and on the practicality of the reported approaches. More specific comments are below.
- Abstract – please grammatically correct the first sentence “receiving attention in the biosensor due…”
- Abstract – it says that they are difficulties in commercializing aptasensors due to performance, however not clear what are the performance problems are.
In addition, throughout the manuscript problems with aptasensors are not discussed. Which contradicts the Abstract.
- Authors should include a summary table of the papers they cited, where the columns can be: type of cytokine(s) detected, limit of detection, sensing method, etc
Such summary table will be very useful to the readers.
- Section “5. Strategies for improving performance” – prior to this section, all previous works have been summarized in a positive way without mentioning drawbacks or limitations. Words like “excellent limit of detection” are used, but no mention of limitations and problems.
The authors did not explain why there is a need to improve performance in the first place.
Therefore a paragraph in the beginning of section 5 should be added explaining why there is a need to improve performance and discuss what type of sensor aspects should be improved and why – detection limits, sensitivity, cost, etc?
- In sections 2,3,4,5 it is important to add some detail of the limitations of each reported sensor in addition to reported performance. Review paper should critically discuss both pros and cons.
For example, line 454 – a study that uses a refractive index to measure cytokine IL-6. Refractive index measurements rely solely on the selectivity of aptamer for detection and have interference from non-specific binding. In the beginning of the paper, the authors say that sensitivity is one of the main problems of aptamers. However in section 5 “Strategies for improving performance”, selectivity is not even mentioned.
Reviewer 3 Report
The authors reviewed the aptamer sensor for cytokines. I would recommend accepting this paper with major revision.
- A table listing the sensor, detecting method, LOD, range etc. will make the information more clearly.
- A content at the beginning is needed.
- Does reference 115 use aptamer (Line 473)?
- “The use of aptamers as receptors is beneficial for electrical detection because….(Line 324)”. “The advantages of aptamer-functionalized FET…(LINE 383)”. More comments like the sentence above are expected. The advantage and disadvantage about different detecting methods should be shown.
- 5. The conclusion is too short. There should be more about the challenge and prospect.
- 6. An editing error, two “was“ were found in Line 204.
Round 2
Reviewer 2 Report
The authors have addressed reviewers' comments.
Reviewer 3 Report
There are still two small questions. After solve these, the manuscript can be accepted.
- Table 1 should contain the probe or sensor, which means the material or strategy used in the reference.
- The content, which means a list of the chapters or sections given at the beginning to show where (in which page) is each section.